# Prevalence of Language Delay among Healthy Preterm Children, Language Outcomes and Predictive Factors

**DOI:** 10.3390/children8040282

**Published:** 2021-04-06

**Authors:** Miguel Pérez-Pereira

**Affiliations:** Department of Developmental and Educational Psychology, Faculty of Psychology, University of Santiago de Compostela, 15705 Santiago de Compostela, Spain; miguel.perez.pereira@usc.es; Tel.: +34-600940125

**Keywords:** preterm children, language delay, predictive factors, language development

## Abstract

Language delay (LD) and its relationship with later language impairment in preterm children is a topic of major concern. Previous studies comparing LD in preterm (PT) and full-term (FT) children were mainly carried out with samples of extremely preterm and very preterm children (sometimes with additional medical problems). Very few of them were longitudinal studies, which is essential to understand developmental relationships between LD and later language impairment. In this study, we compare the prevalence of LD in low-risk preterm children to that of FT children in a longitudinal design ranging from 10 to 60 months of age. We also analyze which variables are related to a higher risk of LD at 22, 30 and 60 months of age. Different language tests were administered to three groups of preterm children of different gestational ages and to one group of full-term children from the ages of 10 to 60 months. ANOVA comparisons between groups and logistic regression analyses to identify possible predictors of language delay at 22, 30 and 60 months of age were performed. The results found indicate that there were practically no differences between gestational age groups. Healthy PT children, therefore, do not have, in general terms, a higher risk of language delay than FT children. Previous language delay and cognitive delay are the strongest and longest-lasting predictors of later language impairment. Other factors, such as a scarce use of gestures at 10 months or male gender, affect early LD at 22 months of age, although their effect disappears as children grow older. Low maternal education appears to have a late effect. Gestational age does not have any significant effect on the appearance of LD.

## 1. Introduction

Preterm children are considered to be an at-risk population, though not all of them share the same percentage of risk. Important differences exist among preterm children in relation to different biomedical factors. One of them, gestational age (GA), also determines whether other factors co-exist. Usually, birth weight (BW) is strongly associated with GA, in such a way that the shorter the GA the lower the BW (with the exception of those children small for GA). Preterm children are classified according to GA into 4 groups [1,2,3]: late preterm children (LPT), who have a GA of 34–36 weeks; moderately preterm (MPT) children, with a GA between 32–33 weeks; very preterm (VPT) children, with a GA between 28–31 weeks; and extremely preterm (EPT) children, with a GA below 28 weeks.

The risk of suffering medical complications increases as GA and BW are lower. EPT and VPT children have a greater probability of being affected by them than LPT and MPT children [4]. The most common medical complications affect lungs (bronchopulmonary dysplasia (BPD), respiratory distress syndrome) and cerebrum (intraventricular hemorrhage (IVH), periventricular leukomalacia (PLM)), with important consequences for children’s development [5].

The study of language development in preterm children has offered controversial results. The majority of studies, as well as a few meta-analyses, have reported that preterm children of different ages tend to show lower results than full-term peers in a diversity of language measures [6,7,8]. Not only do PT children show a smaller vocabulary size than their FT counterparts, but they also show a lower level of grammatical skills than their FT peers in their first years [9,10,11,12].

These studies were mostly carried out with samples of EPT or VPT (or very and extremely low birth weight) children, and some of them did not report and/or use clear exclusion criteria. This means that children with serious biomedical complications or sensory, motor and cognitive handicaps may be unidentified, producing a confounding effect with prematurity on the outcomes. In contrast, a few studies, mostly carried out with healthy preterm children with a variety of GAs, have not found significant differences between PT and FT children in different measures of language development taken at different ages [13,14,15].

The research of the prevalence of late talkers (LT) or language delayed (LD) children in PT as compared to FT children, the language outcomes of these children a few years later and the predictors of LD is the main focus of this study. This research is related to the study of the differences in language development between PT and FT children, but of a slightly different kind.

Late talkers or language delayed children are those children between 18 and 36 months of age who show limited language development as compared to their FT peers, in the absence of neurological damage, environmental deprivation, sensory impairment or cognitive delay. The cut-off criterion used to establish LD varies in different studies, and it depends on the reference age. However, quite commonly, those children below the 10th percentile in language tests are considered to be LD or LT [16]. This criterion is usable for children of different ages in the above-mentioned range, while, in contrast, other criteria, such as to have a productive vocabulary below 50 words and no word combinations is only well suited for the age of 24 months [17,18]. Late talkers may have combined expressive and receptive delays or only expressive delays. The estimates of prevalence of LT or LD children oscillate between 9% and 20% of the population of children aged 24–36 months [15,19]. Some of these children (around 50–70%), called *late bloomers*, catch up to their typically developing (TD) peers by 4–5 years of age [20]. This fact points to the difficulty of predicting later language impairment from early language delay. Early prediction of language impairment or developmental language disorder after 5 years of age is a major focus of concern and developmental follow-up.

Table 1 displays the main findings of those studies which have investigated the prevalence of LD in PT children as compared to FT. In all of them, age correction for prematurity has been applied, with the exception of Lee and Lee [21] in which comparisons were performed using chronological age. Only studies which have identified LD with language specific and appropriate tests were included. For this reason, two studies were not included in Table 1, because the test used is not considered a language ability test, but of verbal intelligence (Wechsler Preschool and Primary Scale of Intelligence) [22,23]. Another two studies which used only partial versions (selection of only a few items) of developmental scales [24,25] were also excluded. Those studies which did not use normative scores (percentile, standard deviation (SD)) to establish cut off criteria were not included either [26].

From Table 1, it is clear that there is a great disparity in the estimates of the percentage of children with language delay. The range of prevalence of LD among PT children goes from 8% to 49%, while that for FT children goes from 0% to 30%. This great variation occurs even though the PT children were of similar GA. Practically all (not that of Do and collaborators [34]) of the samples were of VPT and EPT children, with a mean GA of in between 28 and 30 weeks, with the exception of the study by Wolke and his collaborators [28], in which the EPT children were born below 26 weeks of GA. All the samples were of VPT and EPT children, although in some cases this was termed as extremely low birth weight children (ELBW).

The important variation in the percentages found of children with language delay or language impairment could be related to different factors, such as the age of assessment, the instruments used in each study, the cut-off criteria and the characteristic of the participants, particularly the PT children.

The age of assessment varies greatly among the studies, ranging from below 24 months to 72 months of age, although there are many data for similar ages (namely, at 24 and 48 months of age) which have provided diverging results. It is logical to think that the proportion of LD children may change with age, thus introducing differences in the results of the investigations.

The instruments used are not the same in all the studies, which is comprehensible given the differences in the ages of assessment. In many of them different adaptations of the MacArthur-Bates Communicative Development Inventories (CDI) to different languages were used to assess children of 30 months of age or younger [21,30,31,32,33]. In these cases, lack of agreement in the results indicates that other factors must be responsible for the discrepancy.

The cut-off criteria to identify LD vary among the different studies. In many cases the 10th percentile of the normative sample of the instrument as reference has been used as criterium to determine the limit of LD both for the PT and the FT groups. In other cases, the limit was a certain point of SD in relation to the normative sample. Two investigations, however, when establishing the cut-off point did not use the normative sample of the test as a reference but used the FT control group instead [29,33]. This fact shed some doubts upon the adequacy of the comparisons just in case the participants chosen for the FT group might have had a higher performance not coincident with the norms. In other cases, other cut-off criteria were used (scores below −1 SD, below −1.25 SD, below −1.5 SD, below −2 SD, or below developmental quotient (DQ) 85), which creates classification criteria that are more or less stringent, resulting in lower or higher percentages of LD children, respectively.

Finally, the characteristics of the participants in the different studies may also be a source of variability in the results found. It is certainly true that the PT participants of most of the studies were VPT and EPT children around 28–30 weeks of mean GA. However, the selection criteria changed a lot among the revised studies. Some studies established clear exclusion criteria, which are quite strict, and children with serious biomedical complications were not included in the premature group [30]. In these cases, children with major cerebral damage, such as IVH higher than II or PLM, hydrocephalus, BPD, retinopathy of prematurity, visual or hearing impairment or congenital malformations were excluded. In other cases, the criteria were less strict, and only children with some of these criteria were excluded: congenital abnormalities, chromosomal anomalies, coming from homes where the language of the community was not spoken, admission to the Neonatal Intensive Care Unit (NICU), or mother’s use of alcohol or drugs during pregnancy [29,33] were excluded. Other studies do not offer information on the exclusion criteria (Charolais et al., 2014; Lee and Lee, 2016), which does not guaranty (all the contrary) that the sample of PT children are free of these biomedical hazards (IVH, PLM, BPD, etc.). Two studies directly chose participants who were in the NICU for a long stay [32,34]. Another study [27], which used strict exclusion criteria, however, also included children with BPD in the PT group because one aim of the study was to test the effect of this disease on the risk of suffering language delay. It seems reasonable to think that these differences in the inclusion/exclusion criteria may have important consequences in the differences found regarding the prevalence of language delay. It is enlightening that when Stolt et al. [33] compared only VPT children without neurological damage with FT children, the differences in percentages of LD children are lower, and no significant differences in LD were found in the tasks administered, with the exception of the Nepsy language score at 5 years of age. Complementarily, when VPR children have additional handicaps (such as BPD) they show a higher incidence of language impairment, which may rise to 43% [27,30].

The comparability of the FT and PT groups in certain critical characteristics, such as similarity of maternal education, Socio Economic Status (SES), or balanced gender distribution, is a key point, which is fulfilled by several investigations [27,30,31]. A few studies do not provide information in this regard or not enough information [21,34], and others clearly do not fulfil these requirements, and the PT group is composed of children whose mothers have lower education and/or SES than those of the FT group [5,29]. All these introduce serious doubts on the interpretation of the results found, because the FT and PT groups were not comparable, which introduces a threat to the internal validity of the investigations. On some occasions there was no control group of FT children; instead, the normative sample of a given test was used as the comparison group [32,34].

Only two studies adopted a longitudinal perspective, with repeated measures for the participants [30,33], although in one of them [30] one longitudinal sample of PT children is compared to two different cross-sectional FT samples at different ages (30 and 42 months). This increases the variability of the two samples of FT children, which may differ in their characteristics. Therefore, intraindividual patterns of change cannot be observed for FT children, which is a limit for the accurate longitudinal comparisons of the PT and FT groups.

The results of the control groups are really very unusual in some of the studies. There is always a certain percentage of FT children who are below the cut-off criteria to define language delay (or language impairment for older children), which is usually over 7% [35,36]. However, there are two studies in which the percentage of FT children below percentile 10 or below −2 SD is 0% [21,34]. In the case of Lee and Lee [21] chronological age (not the corrected age) was used for comparisons, and this fact can explain the unusual gap between PT and FT children. In addition, as mentioned before, in these two studies [21,34] no information is provided on the similarities of the PT and FT samples in sociodemographic characteristics (e.g., parental educational level or SES) or gender, which are important characteristics to ascertain that both samples are comparable.

Most of the reviewed studies were conducted with VPR or EPR children, who are considered to be at higher risk of suffering developmental problems than other populations of preterm children, such as moderately or late preterm. In addition, in an important number of the studies carried out, the VPT/EPT children have other associated medical problems (neurological damage, BPD) or risk situations (stay in the NICU for a relatively long time). For this reason, it is not a surprise that the incidence of language delay or language impairment of VPT and EPT children clearly exceeds that of FT children. This population represents around 20% of the total population of PT children [2], which provides a reason to extend studies on the prevalence of LD in PT children to other segments of the total population of PT children in order to get a wider panorama of what happens with the PT population.

Therefore, and this is a purpose of the present study, there is a need to study a sample of PT children with a relatively wide range of GA, and with no serious biomedical complications. On the other hand, there is a dearth of studies carried out with a longitudinal design. One important advantage of longitudinal studies, apart from the description of intraindividual change, is that they allow us to investigate the predictive effect of different factors on the determination of language delay. In this research a longitudinal follow up of 3 groups of PT children with different GAs and one group of FT children will be carried out.

In relation to the most relevant predictive factors of language delay or language impairment, previous research has highlighted a variety of biomedical, environmental and psychological factors, which will be briefly reported on as follows.

Among biomedical factors, gestational age or birth weight were found to have a predictive effect on language delay, and the risk of suffering language impairments [9,10,23,37,38]. Other authors, however, have suggested that neurobehavioral outcomes at an early school age can be predicted based on IVH incidence as opposed to birth weight or GA [39]. Neurological impairment (IVH, PLM), on its own, or in association with other factors has also had an important effect on language delay [5,29,40]. IVH higher than II, but not lower, has been found to have negative effects on cognitive and language measures [41]. Bronchopulmonary Dysplasia seems to have a very detrimental effect on the possibility of PT children having language delay [27,30,31,42,43,44]. Male gender has been found to increase the risk of language delay [28,31,35,36], and family history of language or learning disorders predicted lower language development [15].

Several studies found an influence of environmental factors on language delay such as the level of maternal or parental education [19,20,31,34,36,45,46,47,48], the SES [49], a combination of these two factors [40], and the quality of home environment [50].

Finally, among the psychological or personal factors, previous cognitive development [15,29,31,51,52], previous use of gestures [31,53,54,55], and previous language abilities [30,33,56,57] are good predictors of later linguistic development.

There is a lack of information, however, on the prevalence of language delay in low-risk PR children, and on whether this rate increases as children grow older.

One major strength of the present research is that there is a longitudinal follow up of four groups of children with different gestational ages, covering a range from extremely preterm to full term children (GA 26–41 weeks). Another strong point is that the PT children do not have major medical complications, being considered healthy or low-risk children, a group which paradoxically has been scarcely studied despite that they constitute the majority of newly born PT children.

The aims of the present research are the following:(1)To compare the prevalence of language delay in healthy preterm children (PR) with different GAs to that of full-term children (FT) in a longitudinal design ranging from 10 to 60 months of age.(2)To analyze which variables are related to a higher risk of language delay at 22, 30 and 60 months of age.

## 2. Materials and Methods

### 2.1. Participants

One group of FT and another of PT children were recruited at birth in four hospitals in Galicia (Spain) and longitudinally followed and assessed at different points in time.

The initial participants of the PT group were 151 PT children (with GA range between 26 and 36 weeks), and those children with the following characteristics were excluded: cerebral palsy (as diagnosed up until 9 months of age), periventricular leukomalacia (PLM), intraventricular hemorrhage (IVH) greater than grade II, hydrocephalus, encephalopathy, genetic malformations, chromosomal syndromes and metabolic syndromes associated with mental retardation, important motor or sensorial impairments, and Apgar scores below 6 at 5 min. The initial participants of the FT group were 49 children with standard GA and no evidence of impairment. The children were assessed at 10, 22, 30, 48 and 60 months of age.

The number of participants and their distribution by GA groups at every assessment point is displayed in Table 2. The participants were distributed into four groups according to their GA.

The PT and FT groups did not differ in terms of mother’s education (X^2^ (1) = 8.66, *p* = 0.194), gender (X^2^ (1) = 0.000, *p* = 0.997) or Apgar score (t (197) = −0.909, *p*= 0.365).

The characteristics of the samples remained similar throughout time.

### 2.2. Instruments

To assess language development the Inventario do Desenvolvemento de Habilidades Comunicativas (IDHC) Palabras e xestos (Words and Gestures) and Palabras e Oracións (Words and sentences) [58,59] were filled in by the children’s mothers when the children were 10, 22 and 30 months of age (see below). The IDHC is the Galician version of the MacArthur-Bates Communicative Development Inventories (CDI) [60]. The IDHC Words and Gestures for children between 8 and 15 months of age, was applied when the children were 10 months old. The form Words and Sentences, for children aged between 16 and 30 months was applied when the children were 22 and 30 months of age. The following measures were taken into consideration for the present study. At 10 months of age, word understanding and first communicative gestures, which will be considered as predictive factors in regression analyses. At 22 and 30 months of age, word production, which is considered the central feature identifying language delayed children [51].

When the children were 48 months of age, they were assessed through the Reynell Developmental Language Scales [61]. The RDLS is comprised of two scales: expressive and comprehension language scales. Because of the deficient adaptation of the RDLS into Spanish (no Spanish norms exist, and no adaptation to the characteristics of Spanish language acquisition has been made), only the total raw score in comprehension was used in the analyses performed.

The children’s language development was also assessed when the children were 60 months of age through the Peaboby Picture Vocabulary Test 3rd edition (PPVT-III) [62], the test Comprensión de Estructuras Gramaticales (CEG) [63], and the production scale of the Test de Sintaxis de Aguado (TSA) [64].

The widely known PPVT-III was used to assess vocabulary comprehension. The child is required to point to the image that best matches the word pronounced by the researcher out of the 4 pictures on the page. The words that are tested are arranged in order of increasing difficulty.

The CEG was used to assess the comprehension of syntactic structures. The CEG is a Spanish test that is very similar to the well-known Test of Reception of Grammar (TROG-2) [65]. The CEG consists of 80 pages that include four pictures on each page. In each item, the researcher pronounces a sentence (e.g., “El niño que mira a la niña está comiendo”: The boy who looks at the girl is eating) and the child points to the image that matches the target sentence. The other three images act as (lexical or grammatical) distractors. The CEG explores 20 different syntactic structures organized into blocks with 4 items each. The CEG can be administered to children from 4 to 12 years of age.

The production subscale of the TSA [64] was used to assess morphological and syntactic production skills. In this test, the child has to imitate a sentence previously produced by the researcher looking at a drawing related to the sentence “what did I say about this drawing?”. Thirty items follow this pattern, and in another four items the child has to complete the last part of a sentence given a conversational context created by the researcher (“cuando hace frío….me pongo el abrigo” “when it is cold….I put on my coat”/“si hiciera calor….no me pondría el abrigo” “if it were warm…I would not put on my coat”). The TSA explores the production of different morphosyntactic abilities: interrogative sentences, negative sentences, passives, use of possessive, relative, interrogative, possessive and demonstrative pronouns, complex sentences, comparisons, use of prepositions, use of different persons and times in verbs, etc.). The TSA can be administered to children from 3 to 7 years of age.

The cognitive development of the children was assessed through the Batelle Developmental Inventory (BDI) [66] when they were 22 months of age. This scale measures a child’s progress in development and in discrete skill sets. The skills assessed by the Batelle scale are adaptive, personal-social, communication, motor, and cognitive. The cognitive score was used for the present analyses.

The mothers of the children completed an interview at the beginning of the study that included socio-demographic information of the family, information on pregnancy, Apgar scores, feeding and health habits, educational level of the parents, etc.

The children lived in a bilingual Spanish-Galician community context which makes it possible to use Spanish or Galician tests. The Galician tests (IDHC) were administered to the mothers of the children. The rest of the tests in Spanish were administered to the children. No adaptations of these tests exist for Galician.

### 2.3. Procedure

Previous consent from the mothers was obtained, as well as the acceptance by the Comité Ético de Investigación Clínica de Galicia.

The children’s communicative and linguistic development was assessed at 10, 22, 30, 48 and 60 months of age (±15 days), with corrected age for the PT group up until 30 months of age but not later.

The parent reports (IDHC) were filled in by the mothers. The remaining tests were administered by a trained psychologist at the specified ages in the children’s homes.

The following measures were taken into consideration for the present study. At 10 months of age, word understanding, word production, and first communicative gestures were considered. At 22 and 30 months of age, word production scores were taken into account. We used this measure because at this age word production is the most reliable indicator of language development.

### 2.4. Analyses Performed

The following analyses were performed.

ANOVA for mean comparisons between the results of the PR and FT children at different ages in different measurements.Chi square comparisons between the four GA groups of children and also between the PT and the FT groups regarding the relative percentages of children with and without language delay. Those children with raw scores lower than percentile 10 were considered to have language delay. This criterion, however, was changed in the case of cognitive development [66] at 22 months of age. In this case we have adopted the threshold of percentile 15, because the norms offer percentiles for a range between 18 and 23 months of age, and the children were at the upper limit of the age range.Five logistic regression analyses (enter method) were performed in order to test the predictors of language delay as measured through the different instruments at different ages (dependent variables DV). Previously, the effects of many different variables were tested, and only those which had an effect on the DVs were incorporated in the final analyses, as well as 3 variables of theoretical relevance: gestational age (numerical), gender and maternal education level (three groups: low, medium and high). Among those variables which did not have any effect on any DV were: Apgar score in the 1st minute (risk/no risk = ≥7)), stay in the NICU (1 = no stay, 2 = 1–15 days, 3 = >15 days), family antecedents of language problems (yes/no), mother’s age at birth (risk/no risk), risk of maternal depression (yes/no), parental stress (risk/no risk), HOME score (quality of home environment). In addition, the absence of effect of some of them on language risk/delay has been demonstrated in a preliminary research [67]. These variables were not included in the regression models, and, therefore, no information is offered on them for brevity’s sake.

The logistic regression analyses were carried out with all the participants, because the number of FT children was not large enough to perform separate analyses for PT and FT children, and for the sake of the strength of the tests.

In the first logistic regression model, the dependent variable (DV) was children with or without lexical delay (word production) at 22 months of age. The predictive variables introduced were: gestational age in weeks, gender, maternal education, total score in first communicative gestures at 10 months (IDHC) and total score in vocabulary understanding at 10 months of age (IDHC).

In the second logistic regression model, the dependent variable (DV) was children with or without lexical delay at 30 months and the Predictors were those factors which in previous logistic regression analyses had had a significant effect on the DV or theoretical relevance: gestational age, gender, maternal education, risk of cognitive delay at 22 months (BDI), and risk of vocabulary delay (word production) at 22 months of age (IDHC).

In the following three logistic regression analyses the Dependent variables were children with or without language delay at 60 months of age. The threshold was percentile 10 in the the PPVT-III (vocabulary comprehension), CEG (grammar understanding), and the TSA (morphosyntactic production) in each analysis. The predictive variables were always the same for these three analyses. The Predictors were those factors which in previous logistic regression analyses had had a significant effect on the DV or theoretical relevance: gestational age, gender, maternal education, risk of cognitive delay at 22 months, risk of vocabulary delay at 30 months, and total score in the comprehensions scale of the RDLS at 48 months of age.

## 3. Results

### 3.1. Descriptive Results and Comparisons between Groups

The results of the one factor ANOVA to compare the GA groups in different measures is offered in Table 3. As can be observed, there were no significant differences between the groups in any measure, with the exception of the CEG at 60 months of age (*p* < 0.05). In this case, the differences were due to the differences between the GA ≤ 37 week group and the GA 36–34 week group (Bonferroni post hoc *p* < 0.05).

### 3.2. Language Delay Comparisons between Groups

The percentages of children in each GA group with language delay (LD) as assessed through different measures taken at different ages are presented in Table 4. This table indicates the number and relative percentage of children of each GA group who got a score below percentile 10 in the tests applied at different ages. Those children are considered as part of the language delay/language impairment group. In addition, the results of the chi squared test are also presented.

In general terms, there are no significant differences in the proportion of children with language delay/impairment among the four groups, with the exception of the results with the CEG (grammar comprehension) (X^2^ = 9.378, *p* < 0.05), in which the GA groups 36–34 and 33–32 weeks clearly have a very high percentage of children with LD (40.5% and 38.7%, respectively), while the groups with children having a GA of 37 weeks or above and with a GA of 31 weeks or below have lower (and quite similar) proportions of children with LD (15.2% and 17.6%, respectively).

In order to make the results more manageable and to make the comparisons clearer, we have integrated the results of all the GA groups below 37 weeks in a group of preterm infants. These results are presented in Table 5.

In this case, the difference between the PT and the FT groups in the CEG is only marginally significant (X^2^ = 3.810, *p* = 0.051). The rest of the comparisons do not reach significance. There are no significant differences between FT and PT children in vocabulary production at 22 and 30 months of age. Nor there are differences in receptive vocabulary (PPVT-III) or morphosyntactic production (TSA). The frequency of children with delay in the PPVT-III is clearly lower than in the rest of the tests, although the percentage of PT (6.4%) children with LD is double the percentage of FT (3.0%) children.

Table 5 also displays the percentage of PT and FT children with cognitive delay measured at 22 months of age through the BDI, because this score will be used in logistic regression analyses. No significant difference between FT and PT children is found in this regard.

### 3.3. Logistic Regression Analyses

The following tables display the results of the logistic regression analyses performed.

In Table 6, the results of the logistic regression for delay/not delay in word production at 22 months of age (IDHC Words and Sentences) as the dependent variable are presented. Out of the predictors introduced in the model, only the total score of first communicative gestures at 10 months of age (*p* = 0.013), gender (*p* = 0.032) and the total score in word comprehension at 10 months of age (*p* = 0.046), in this order, were found to have a significant effect on the variance of having or not having language delay as measured through word production at 22 months of age. The variance explained by the model is moderate (Negalkerkes’s R^2^ = 0.153). The model reaches significance (Hosmer– Lemeshow’s X^2^ (8) = 9162, *p* > 0.329; *X^2^* (6) = 19.348, *p* = 0.002; −2LL = 178.578), and correctly classifies 78.9% of the participants (specificity: 97.8, sensitivity = 18.6).

In Table 7, the results of the logistic regression for delay in word production at 30 months of age (IDHC Words and Sentences) as the dependent variable are presented. The only significant predictors found are risk of vocabulary delay (*p* = 0.000) and risk of cognitive delay (*p* = 0.038) at 22 months of age. The variance explained by the model is 34% (Negalkerkes’s R^2^ = 0.344). The model reaches significance (Hosmer–Lemeshow’s *X^2^* (8) = 9005, *p* > 0.342; (*X^2^* (6) = 36.341, *p* = 0.000; −2LL = 109.292). The model correctly classified 83.7% of the participants (specificity: 95.2, sensitivity = 32.7).

Table 8 shows the results of the logistic regression analysis for delay in lexical comprehension (PPVT-III) as the dependent variable. In this case, the analysis has to be interpreted with caution, since the frequency of those children with scores under percentile 10 are only 8. The predictive variables which have a significant effect are risk of vocabulary delay at 30 months of age (IDHC-Words and Sentences) (*p* = 0.022) and the total score in language comprehension (RDLS) at 48 months of age (*p* = 0.046). Maternal education has a nearly significant effect (*p* = 0.056). The variance explained by the model reaches 41% (Negalkerke’s R^2^ = 0.414). The model reaches significance (Hosmer–Lemeshow’s *X^2^* (8) = 9160, *p* = 0.329; *X^2^* (6) = 18.143, *p* = 0.006; −2LL = 30.765). The model correctly classified 96.2% of the participants (specificity: 99.2, sensitivity = 33.3).

In Table 9, the results of the logistic regression for delay in syntactic understanding (CEG) at 60 months of age as the dependent variable are presented. The only significant predictors found are total score in language comprehension (RDLS) at 48 months of age (*p* = 0.009) and risk of cognitive delay at 22 months of age (*p* = 0.012). The variance explained by the model is 27% (Negalkerkes’s R^2^ = 0.278). The model reaches significance (Hosmer–Lemeshow’s *X^2^* (8) = 4793, *p* = 0.779; *X^2^* (6) = 27.853, *p* = 0.000; −2LL = 124.838). The model correctly classified 80.3% of the participants (specificity: 96.9, sensitivity = 34.3).

Finally, Table 10 shows the results of the logistic regression analysis for delay in morphosyntactic production (TSA) at 60 months of age as the dependent variable. The predictors which reach significance are the total score in language comprehension (RDLS) at 48 months of age (*p* = 0.003) and risk of vocabulary delay at 30 months of age (*p* = 0.042). The model explains 17% of the variance (Negalkerke’s R^2^ = 0.176). The model reaches significance (Hosmer–Lemeshow’s *X^2^* (8) = 4671, *p* = 0.792; *X^2^* (6) = 17.472, *p* = 0.008; −2LL = 145.351). The model correctly classified 69.5% of the participants (specificity: 91.1, sensitivity = 22.0).

## 4. Discussion

In relation to the first aim of the study which was to compare the prevalence of language delay in healthy preterm children (PR) with different GAs to that of full-term children (FT), the results found indicate that there are no significant differences in the percentage of children with language delay among the four GA groups in the following language measures: Word production at 22 and 30 months of age as measured through the Galician CDI, word comprehension at 60 months of age as measured through the PPVT, morphosyntactic production at 60 months of age as measured through the TSA. The only significant difference was found in grammatical structures comprehension (*p* < 0.025), measured through the CEG. The greatest differences occurred between the GA groups of 36–34 and 33–32 weeks (with 40.5% and 38.7% of LD respectively) and the other two groups (FT and VPT-EPT, with 15.2% and 17.6%, respectively). This result is coincident with that found in the ANOVA (Table 3), in which the significance was explained by the difference between the groups GA ≥ 37 and GA 36–34 weeks. Unexpectedly, the difference was not due to the difference between the most distant groups (≥31 and ≥37 weeks), that is to say the VPT-EPT and the FT groups. Therefore, the GA factor does not seem to explain these results, contrary to other authors’ claims [9,10,37]. This conclusion will be confirmed later with the regression analyses and should be interpreted taking into consideration the low-risk characteristic of the sample.

When the results of all the PT children (GA < 37) are put together, the comparison is simpler and, again, the results indicate no significant differences in the language measures taken. Even in the test of comprehension of grammatical structures (CEG), administered at the age of 60 months, the difference in this case does not reach significance, although it is really very close (*p* = 0.051).

In general terms, the percentage of FT children with LD throughout time, using different tests, remains quite stable (with the exception of the PPVT-III results) in a range between 15.2% and 21.9%. In contrast, the percentages of children with LD in the PT group vary much more over time, in a range between 17.9% and 33%, and there is not a clear incremental trend in the percentage of children with language delay from early years to 5 years of age, as several authors have proposed [30,33,37]. It is obvious that using different tests with different norms makes comparisons throughout time difficult to carry out because variability can be caused not only by changes in the participant, but also by variations in the norming process. Therefore, the results must be taken with caution.

One factor that seems to increase the risk of undergoing language delay in PR children is the existence of medical complications (neurological or pulmonary) [27,33]. When these children were excluded, the rate of language delay of the PR children descended. Probably, the fact that our sample was practically free of children with these medical complications may have affected the results found in the PR group. One additional argument in favor of this idea is that those investigations which included relatively high percentages of PR children with neurological or pulmonary medical problems evidenced very high rates of language delay for PR children [27,29,42,43].

In relation with the second aim, which was to identify those variables related to a higher risk of language delay at 22, 30 and 60 months of age, the results found in the logistic regression analyses permit identification of different predictive factors, which vary according to the moment of assessment as well as the different linguistic abilities.

Three factors were found to have an effect on the probability of suffering from language (lexical) delay at 22 months of age (Table 6): Gender, use of first gestures at 10 months, and total vocabulary understanding at 10 months. Gender reached significance (*p* < 0.05, OR = 2.312), with boys having a higher risk of language delay than girls (more than twice as high). This result is in agreement with other studies with PT and FT children of a similar age [11,13,31] and older [28,35], and does not support the results found by other studies [15] which found practically no effect of gender on word production in children of 30 months of age, or at 24 and 60 months of age [33].

The use of first gestures (total score) also had a significant effect on language delay at 22 months of age (*p* < 0.05, OR = 0.820), indicating that those children with a lower number of gestures at 10 months of age have a higher probability of being language delayed at 22 months of age, which agrees with former investigations carried out with children of similar ages [31,53,55]. Therefore, this result confirms that the use of gestures seems to be a possible predictor of language development in the short term.

Finally, the third factor which has been found to have a significant effect on lexical delay at 22 months of age was word comprehension at 10 months of age (*p* < 0.05, OR = 1.005). In any case, this effect was very reduced (OR = 1.005), and apparently paradoxical (see 33) and contrary to expectations.

In general terms, the logistic regression model for word production at 22 months of age correctly classifies 79% of the participants, even though the sensitivity is low; this means the classification of the participants in the delayed group is not good, with a high proportion of false negatives (children who are not classified as language delayed although they are language delayed).

In relation to the prediction of language delay at 30 months of age (Table 7), two risk factors seem to have a significant effect: cognitive delay at 22 months and productive vocabulary delay at 22 months of age. Vocabulary delay at 22 months has an important impact on later language (lexical) delay (*p* < 0.001, OR = 8.712), indicating that those children with lexical delay at 22 months of age have many more possibilities of suffering from language delay at 30 months of age. Cognitive delay at 22 months also has a significant (although somewhat lower) effect on the probability of suffering language delay at 30 months (*p* < 0.05; OR = 3.386). These results are in tune with those found in other studies which have claimed that previous linguistic [30,33,57] and cognitive development [11,15,29,51,52] are good predictors of later language delay.

The model correctly classifies 84% of the participants in the two groups of language delayed and not language delayed, with a high specificity (95.2) even though the sensitivity is low (32.1), indicating that, again, there is a high percentage of false negatives.

Other factors which were found to have a significant effect on early language delay, such as gestational age [9,10,23,37], or maternal education [19,20,33,34,36,45,46,47,48] did not reach significance at either 22 or 30 months of age. Other authors [15], however, found that low parental education level quite unexpectedly did not affect child linguistic outcomes at the age of 36 months.

The logistic regression analyses performed when the participants were 60 months old, give interesting results which point to the effect of previous language, cognitive delay and maternal education level.

The results obtained in the regression analysis with vocabulary comprehension (PPVT-III) at 60 months of age as dependent variable must be taken with caution, because of the low number of children who scored below percentile 10 (8 in all). In this case, those children who were language delayed (vocabulary production) at 30 months of age have a much greater probability of being language delayed (receptive vocabulary) 30 months later (*p* < 0.05, OR = 19.172). Those children who got low scores in language comprehension (RDLS) at 48 months of age have also got a greater probability of being in the group of language delayed children (receptive vocabulary) at 60 months of age (*p* < 0.05 OR = 0.848).

The model correctly classifies 96.2% of the participants in the two groups of language delayed and not delayed, with a high specificity (99.2) but a modest sensitivity (33.3), indicating that there is a high percentage of false negatives.

Three predictive factors seem to be involved in grammar understanding at 60 months of age (CEG): maternal education level, cognitive delay at 22 months, and language comprehension at 48 months of age. Low maternal education increases the probability of having children with language impairment (grammar understanding) at 60 months (*p* < 0.01, OR = 0.434). It is interesting to note that maternal education at this point has a significant effect, but this effect did not exist when the participants were younger. This apparently points to a cumulative effect of maternal education level throughout time, which is compatible with Linsell’s et al. [68] suggestion that the impact of environmental factors on cognitive development becomes more prominent over time for VPT children. Other authors suggested the same cumulative effect for language development [14,31,69,70]. Another interesting and somewhat surprising result is that cognitive delay measured at 22 months of age still has a predictive effect on grammar understanding impairment at 50 months of age (*p* < 0.05, OR 5.929), demonstrating a long lasting and strong (OR value) effect, which is coincident with other findings for VPT children of the same age [33]. Not surprisingly, low scores in understanding language (RDLS) at 48 months increment the possibility of having delays in grammar understanding at 60 months of age (*p* < 0.01, OR = 0.891), remarking the predictive role of previous language abilities in the same domain (understanding).

The model (Table 9) correctly classified 80.3% of the children into the two categories of language delayed and not delayed, and has a high specificity (96.9), although a relatively low sensitivity (34.3).

Finally, two factors seem to have a significant predictive effect on morphosyntactic production delay at 60 months of age: vocabulary delay at 30 months, and low scores in the language comprehension scale of the RDLS. The fact of having language (lexical) delay at 30 months increases the probability of having morphosyntactic impairments 30 months later (*p* < 0.05, OR = 2.977). Those children with low scores in language understanding (RDLS) also have a higher probability of being delayed in morphosyntactic production one year later (*p* < 0.01, OR = 0.880).

This time the model (Table 10) is less powerful in the process of classifying the children into the two groups (language delayed/not language delayed in morphosyntactic production) since only 69.5% of the children are correctly classified. Although specificity is high (91.1), sensitivity is even lower than in the other regression analyses (22.0), thus indicating the existence of many false negatives.

The use of a longitudinal design, in which the children were followed from 10 to 60 months of age, allows for the revelation of certain findings which would not be patent in a cross-sectional design or a short-term longitudinal design.

First, these results show that certain predictors of early language delay (22 months), such as a low number of gestures produced at 10 months of age or low vocabulary understanding at the same age, lose their effect as children grow older, contradicting the results of other studies [54]. Similarly, gender seems to have an effect on language delay at the beginning (22 months of age), while it seems to lose its effect on later language development [15,33].

Complementarily, low maternal education does not have an effect on language delay during the first stages of language development (22 and 30 months of age) but, however, emerges as a predictive factor of grammar understanding delay at the age of 60 months. This pattern shows that environmental factors have a cumulative or incremental effect on language development over time [14,31,69,70]. In any case, the effect of maternal education is not general throughout all the linguistic domains. Possibly, grammar understanding assessed through the CEG is more demanding on abilities linked to the effect of family activities and cultural practices which are developed in families with mothers who have a medium to high educational level than the other tests are (PPVT-III, or TSA).

Another surprising (and relatively unexpected) result is the long-lasting effect of cognitive delay on language delay measured at different ages (30 and 60 months). This reinforces the idea that cognitive development is one of the most powerful predictors of language development [14,15,31,51,52], particularly for PT children. Again, the effect of cognitive delay is more evident in the case of grammar understanding, probably because this test is more demanding of cognitive resources (including working memory) than the other tests used.

Previous language delay has also been found to have an important effect on later language delay [14,30,33,57]. This occurs particularly if the domains of language measured are linked and if the time spent between the ages of measurement is not very long.

The different models tested in the logistic regression analyses can only explain a relatively modest percentage of the variance in the different linguistic measurements, ranging from Negalkerke’s *R^2^* = 0.153 in the case of word production at 22 months of age to Negalkerke’s *R^2^* = 0.414 in the case of vocabulary comprehension at 60 months of age. This indicates that other factors, whose effects have not been studied in this research, may also be predictors of language delay at different ages. In fact, the low sensitivity values found (ranging from 18.6 to 34.3) gives support to the former idea.

As a final conclusion, but not less important, GA does not seem to have any important effect on the prediction of language delay in the case of healthy preterm children when no serious handicap is associated, coinciding with other research findings [14,15,71]. These results of the regression analyses also agree with the ANOVA comparisons, giving strength to the conclusion. The fact that gestational age did not have any significant effect on the language delay of low-risk PT children needs to be highlighted, since this is a novel result in the literature, and it contrasts with other previous studies carried out with VPR or EPR children [29,30,33]. Again, this conclusion has to be taken with caution and cannot be generalized for PT children with other conditions.

## 5. Conclusions

Several conclusions can be drawn from this study. First, healthy PT children do not have, in general terms, a higher risk of language delay than FT children, and seem to have a lower risk of language delay/impairment than very preterm or extremely preterm children studied in other investigations. Second, previous language delay and cognitive delay are the strongest and longest-lasting predictors of later language impairment. The effect of certain predictors of early language delay, such as a low number of gestures and low vocabulary understanding at 10 months of age as well as gender, disappears as language development evolves. On the contrary low maternal education affects language delay after a certain point, indicating a cumulative effect over time.

A limitation of this study is that the effect of medical problems on PT children’s language delay could not be studied since children with medical problems were excluded.

## Figures and Tables

**Table 1 children-08-00282-t001:** Summary of the investigations which compared the prevalence of LD in preterm (PT) and full-term (FT) children.

References (Ordered by Publication Year)	FT and PT Group Characteristics	Age of Assessment	% of Delayed PT Children	% of Delayed FT Children	Language Measure and Classification Criteria
Singer et al. [27]	98 VLBW with BPD Mean GA 27 (SD:2) weeks.	36 months			BDI communication subscale
	70 VLBW without BPD. Mean GA: 30 (2)				DQ < 85 (equivalent to <−1 SD)
	95 FT children. Groups did not differ in gender, maternal education SES, race. Exclusion: serious neurological problems, socially disadvantaged.		49%BPD/34%	30% ^$^	Receptive
			44%BPD/25%	25% ^$^	Expressive
			43%BDP/31%	28%	Overall
Wolke et al. [28]	241 EPT children (GA ≤25 weeks)	72 months			PLS-3 < −2 SD
			15.6%	1.9% *	Total score
			9.5%	1.3%	Receptive
			12.1%	1.3%	Expressive
Woodward et al. [5]	105 EPT/VPT Mean GA: 28 weeks. 16%: cerebral palsy	48 months			CELF-P < −1 SD
			30%	15.2%*	Receptive
	107 FT with higher SES than the PT		25%	12.4%*	Expressive
			31%	15.2%*	Overall
Foster Cohen et al. [29]	105 VPT (stay in NICU): 19 with moderate or severe and 60 with mild white matter abnormality. Mean GA: 27.8 (2.3)	48 months			CELF-P
	107 FT children				Overall
	Exclusion: congenital		16%	8.6%	Mild LD: >−1 SD below the mean of the FT group
	abnormalities and non-English speaking parents.		15%	6.7%*	Moderate or severe LD: <−1,5 SD
	SES and maternal education of FT significantly higher.		31%	15.3%	Total LD
Sansavini et al. [30]	64 very preterm. Mean GA: 30.4 (2.1) without serious complications longitudinally followed.	30 months	24.1% (CDI)	13.6%	Italian CDI total word prod. 10th percentile
			16.1% (MLU)	9.1%	PRF MLU < −1.25 SD
	FT group: 22 / 40 (at age 30 and 42 months)	42 months	34.4% (MLU)	7.5% *	PRF MLU < −1.25 SD
Sansavini et. al. [31]	104 VPT children.	24 months	20%	5% *	Italian CDI short form word product.
	Mean GA: 29.5 (2.1) weeks 20 FT children				<10th percentile
Charollais et al. [32]	117 VPT children GA range: 25–32 weeks. Stay in NICU. No medical or demographic information.	24 months	41%	10% *	French CDI short form. Word production.
	Control: Normative sample, N = 385				<10th percentile.
Stolt et al. [33]	141 VPT. Mean GA: 28 (3) weeks. 146 FT longitudinally followed	24 months			Finnish CDI
			18%	9%*	Word prod.
			18%	8%*	MLU3
			16%	10%	Bayley-II expressive language
	120–131 VPT 123–137 FT Similar maternal education	60 months	27%	10% *	Nepsy lang.
			20%	10% *	5 to 15 lang scores.
					<10th percentile of the FT group
Lee and Lee [21]	86 PT Korean children. Mean GA = 30.45 weeks. Median stay at NICU: 34 days.	10 to 30 months			Korean CDI word prod.
			46.5%	0% *	word prod
					SELSI:
			43%	0% *	Combined
			50%	0% *	Expressive
			34.9%	0%	Receptive
					10th percentile
Do et al. [34]	184 Vietnamese PT children. Mean GA = 31.6 (2.5).	24 months	8%	0% *	BSID-III: Language Composite Score.
	Mean stay in NICU of 34 days.				<−2 SD
	Control PT: normative sample of 78 children (no sociodemographic information)				

Notes: * Significant differences between delay percentages of the two groups. ^$^ Significant differences between delay percentages of the PT group with BPD and the other two (FT and PT without BPD). BDI: Batelle Developmental Inventories. BSID-III: Bayley Scales of Infant Development-III. CELF-P: Clinical Evaluation of Language Fundamentals- Preschool. CDI: Communication Development Inventories-MacArthur-Bates scales. MLU3: Mean Length of Utterances of the three longest utterances (obtained with the CDI). NICU: Neonatal Intensive Care Unit. PLS-3: Preschool Language Scale-3 (UK), which comprises Auditory Comprehension and Expressive Communication scales. PRF MLU: Prova di Repetizione di Frasi (Italian Test of sentence repetition) Mean Length of Utterances. SELSI: Sequenced Language Scale for Infant. 5 to 15 language: Five to Fifteen Language.

**Table 2 children-08-00282-t002:** Composition of the sample throughout time.

Age	GA ≥ 37 (%)	GA 36–34 (%)	GA 33–32 (%)	GA ≤ 31 (%)
15 days	49	65	37	49
10 months	49	65	37	49
22 months	43	58	36	43
30 months	37	48	32	37
48 months	34	42	33 *	36
60 months	33	42	31	34

Note: * One child not tested at 30 months was tested at 48 months of age.

**Table 3 children-08-00282-t003:** Mean (SD) scores and ANOVA comparisons between the four gestational age (GA) groups.

Task (Age)	GA ≥ 37 Mean (SD)	GA36–34 Mean (SD)	GA33–32 Mean (SD)	GA ≤ 31 Mean (SD)	F	df	*p*
First Gestures (10 months)	7.5 (2.5)	7.2 (2.4)	7.4 (2.7)	6.5 (2.7)	1.262	190	0.289
Comprehension of words (10 months)	71.8 (58.8)	88 (77.2)	71.5 (70)	73.3 (73.3)	0.694	190	0.557
Cognition BDI (22 months)	27.5 (4)	26.7 (3.7)	26.8 (3.2)	26.5 (2.9)	0.743	180	0.528
Word Production (22 months)	173.7 (137.1)	174.5 (163.8)	154.2 (130.1)	140.9 (137.8)	0.573	179	0.633
Word Production (30 months)	411.4 (171.3)	412.58 (189.7)	431.00 (149.2)	408.05 (181.5)	0.116	153	0.951
Comprehension RDLS (48 months)	46.5 (5.2)	43.1 (8.6)	43 (4.7)	44.2 (5.9)	2.230	144	0.087
PPVT (60 months)	62 (12.5)	57.8 (11.4)	57 (12.1)	56.2 (13)	1.460	141	0.228
CEG (60 months)	52 (7.3)	44.4 (13.8)	47.1 (9.1)	48.6 (13.4)	2.804	139	0.042
TSA Production (60 months)	43.5 (8.1)	38.4 (15.8)	41.5 (11.2)	39.7 (15.9)	1.034 *	116,539	0.380

Note: * Brown–Forsythe test; F = value of F-statistic; df = degrees of freedom; *p* = significance value.

**Table 4 children-08-00282-t004:** Frequency and (percentage) of children with language delay at 22, 30 and 60 months of age (<10th percentile), and GA group comparisons.

Assessment (Age)	GA ≥ 37 (%)	GA 36–34 (%)	GA 33–32 (%)	GA ≤ 31 (%)	X^2^	*p*
Word Production (22 months)	8 (18.6)	15 (25.9)	6 (16.7)	14 (32.6)	3.595	0.309
Word Production (30 months)	7 (18.9)	8 (16.7)	4 (12.5)	9 (24.3)	1.720	0.632
PPVT (60 months)	1 (3)	1 (2.4)	2 (6.3)	4 (11.4)	3.490	0.322
CEG (60 months)	5 (15.2)	17 (40.5)	12 (38.7)	6 (17.6)	9.378	0.025
TSA Production (60 months)	7 (16.7)	15 (35.7)	12 (28.6)	8 (19.0)	3.608	0.307

X^2^ = Chi square value; *p* = significance value.

**Table 5 children-08-00282-t005:** Frequency and (percentage) of PT and FT children with language delay at 22, 30 and 60 months of age (<10th percentile), and with of cognitive delay (<15th percentile) at 22 months of age and comparisons between groups.

Risk of Delay	FT (%)	PT (%)	X^2^	*p*
Word Production (22 months of age)	8 (18.6)	35 (25.5)	0.868	0.352
Word Production (30 months of age)	7 (18.9)	21 (17.9)	0.018	0.894
PPVT (60 months of age)	1 (3.0)	7 (6.4)	0.548	0.459
CEG (60 months of age)	5 (15.2)	35 (32.7)	3.810	0.051
TSA Production (60 months of age)	7 (21.9)	35 (33.0)	1.442	0.230
(BDI) Cognitive delay (22 months of age)	6 (14.0)	22 (15.9)	0.099	0.753

X^2^ = Chi square value; *p* = significance value.

**Table 6 children-08-00282-t006:** Logistic regression analysis: predictors of language delay (LD) in word production at 22 months (IDHC).

Variables	B	SE	Wald’s *X^2^*	*p*	OR	95% CI
Gestational Age	−0.057	0.051	1.244	0.265	0.945	0.856–1.044
Gender	0.838	0.391	4.593	0.032	2.312	1.074–4.975
Maternal education	−0.420	0.238	3.130	0.077	0.657	0.412–1.046
Total first gestures 10 months	−0.198	0.080	6.148	0.013	0.820	0.701–0.959
Total vocabulary understanding 10 months	0.005	0.003	3.988	0.046	1.005	1.000–1.011

B = Unstandardized regression weight; SE = Standard error for the unstandardized B; *p* = Significance value; OR = Odds ratio; 95% CI = Confidence interval of the odds ratio.

**Table 7 children-08-00282-t007:** Logistic regression analysis: predictors of LD in word production at 30 months of age.

Variables	*B*	SE	Wald’s *X^2^*	*p*	OR	95% CI
Gestational Age	0.041	0.064	0.408	0.523	1.042	0.919–1.180
Gender	0.350	0.526	0.444	0.505	1.420	0.506–3.980
Maternal education	−0.449	0.326	1.899	0.168	0.638	0.337–1.209
Cognitive delay 22 months	1.220	0.589	4.284	0.038	3.386	1.067–10.746
Vocabulary delay 22 months	2.165	0.512	17.888	0.000	8.712	3.195–23.754

B = Unstandardized regression weight; SE = Standard error for the unstandardized B; *p* = Significance value; OR = Odds ratio; 95% CI = Confidence interval of the odds ratio.

**Table 8 children-08-00282-t008:** Logistic regression analysis: predictors of vocabulary comprehension delay (PPVT-III) at 60 months of age.

Variables	*B*	SE	Wald’s *X^2^*	*p*	OR	95% CI
Gestational Age	−0.231	0.172	1.789	0.181	0.794	0.566–1.113
Gender	−1.344	1.300	1.069	0.301	0.261	0.020–3.332
Maternal education	2.046	1.072	3.642	0.056	7.737	0.946–63.253
Cognitive delay 22 months	0.775	1.622	0.228	0.633	2.171	0.090–52.112
Vocabulary delay 30 m.	2.953	1.289	5.247	0.022	19.172	1.532–239.988
Total comprehension score RDLS	−0.165	0.083	3.969	0.046	0.848	0.721–0.997

B = Unstandardized regression weight; SE = Standard error for the unstandardized B; *p* = Significance value; OR = Odds ratio; 95% CI = Confidence interval of the odds ratio.

**Table 9 children-08-00282-t009:** Logistic regression analysis: predictors of grammar understanding delay (CEG) at 60 months of age.

Variables	*B*	SE	Wald’s *X^2^*	*p*	OR	95% CI
estational Age	−0.015	0.063	0.059	0.808	0.985	0.871–1.114
Gender	−0.363	0.467	0.603	0.437	0.696	0.278–1.738
Maternal education	−0.835	0.314	7.061	0.008	0.434	0.234–0.803
Cognitive delay 22 months	1.780	0.705	6.375	0.012	5.929	1.489–23.608
Vocabulary delay 30 months	−0.040	0.633	0.004	0.950	0.961	0.278–3.323
Total comprehension score RDLS	−0.116	0.044	6.905	0.009	0.891	0.817–0.971

B = Unstandardized regression weight; SE = Standard error for the unstandardized B; *p* = Significance value; OR = Odds ratio; 95% CI = Confidence interval of the odds ratio.

**Table 10 children-08-00282-t010:** Logistic regression analysis: predictors of morphosyntactic production (TSA) delay at 60 months of age.

Variables	*B*	SE	Wald’s *X^2^*	*p*	OR	95% CI
Gestational Age	−0.001	0.057	0.000	0.983	0.999	0.892–1.118
Gender	0.249	0.421	0.351	0.554	1.283	0.563–2.925
Maternal education	0.101	0.281	0.130	0.719	1.106	0.638–1.917
Cognitive delay 22 m	−0.209	0.658	0.101	0.750	0.811	0.223–2.946
Vocabulary delay 30 m	1.091	0.536	4.148	0.042	2977	1.042–8.508
Total comprehension score RDLS	−0.128	0.043	9.095	0.003	0.880	0.809–0.956

B = Unstandardized regression weight; SE = Standard error for the unstandardized B; *p* = Significance value; OR = Odds ratio; 95% CI = Confidence interval of the odds ratio.

## Data Availability

Data supporting reported results can be asked for to the author.

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
