# Peer review of "Prevalence of Language Delay among Healthy Preterm Children, Language Outcomes and Predictive Factors"

_children, 2021, doi:10.3390/children8040282_

Round 1

Reviewer 1 Report

This is a welcome addition to the literature on how premature birth affects language outcomes. Some excellent aspects of the paper are its longitudinal approach, its focus on children who are low-risk, and its use of an array of language measures. The results indicated that on the vast majority of measures across most ages studied, there was no evidence of higher risk of language delay in groups of children born prematurely. I have a few concerns about the paper, however.

While I understand that the authors wanted to control for neurological deficits in their data, it seems that this decision resulted in recruiting a very atypical group of children at the younger edge of GA – that is, children born 25 weeks premature are likely to present with neurological damage, and making it an exclusionary criterion means that the authors recruited a very healthy sample of children born prematurely. Addressing this in the discussion and cautioning the interpretation in light of this would be helpful.

I question the authors’ decision to define language delay in terms of percent of children in each group performing below the 10th percentile on each of the language measures (realizing that this is a frequently-taken approach). On the one hand, this is a rather strict approach, and in many cases, language delay is defined less strictly as scoring 1.25-1.5 SDs below the mean/norm, especially for older children. On the other hand, even with the rather strict criterion of scoring below the 10th percentile, the numbers of children identified as having language delay are somewhat high (19% of children in the group of children born full-term at 22 months, for example, seem to exceed the estimates of late-talking in the population). Could the measures be over-identifying children with language delay? What are the sensitivity estimates for the measures chosen by the authors? Does the low sensitivity yielded by the regression analyses reflect the low sensitivity of the language measures?

I would encourage the authors to consider an alternative (or an additional) approach to their data and analyze the actual scores on the language measures across the groups. Would there still not be differences among groups (or predictive relationships in the regressions with gestational age) if each participant’s actual score on the measure served as the dependent variable? While this does not get at the question of prevalence of language delay, I am not convinced that questions of prevalence can in fact be asked in samples as small as those in the present study.

Given the paradoxical findings, the discussion needs to integrate more consideration of why there was not an effect of gestational age, and why there were not more differences observed between children born preterm and full-term.

Author Response

RESPONSE TO REVIEWER 1

Thank you very much for your comments.

In relation to your first comment, the sample of PT children had a mean GA of 32.6 weeks (SD: 2.43, range 26-36) (I noticed a mistake in the paper, the lowest range is not 25 but 26 weeks). Therefore, the sample was quite representative of the population of PT children. It is well known that the percentage of PT children with serious medical problems increases as GA and birth weight are lower. As far as I know, there are no general data of the PT children without serious medical problems (neurological or pulmonary damage, etc.), in other words, healthy PT children.

It is clear that the characteristics of the sample affect the results found. This is recognized on page 13, lines 516-523, and again on page 16 lines 659-662 of the revised document. In any case, and following your suggestion, I have included a new paragraph at the end of the discussion section (lines 666-667 of the revised version).

In relation to the second comment, the point of the differences in the results because of using different cut-off criteria is dealt with at several points in the paper (lines 69-74, lines 116-127). In any case, the use of the 10thpercentile is quite common, and, by the way, very close to -1.25 SD. As indicated in the text, the prevalence of LD oscillates between 9% and 20% of the population of children aged 24-36 months, and it is not unusual to find percentages around 17%. Obviously, if I had chosen the cut-off criteria of -1.5 or -2 SD the percentage of LD children would be lower. In any case, the important point in relation to the replicability of the result is to define one criterium with clarity.  Regarding the point of sensitivity, I think that the sensitivity of the logistic regression models has to do with the capacity of the models to correctly classify those children with and without language delay. The low sensitivity found is probably related to the fact that other variables not included in the models might have affected the sensitivity scores, as indicated in the text. The tests used are commonly recognized tasks, although I do not know of studies carried out to test their capacity to identify language delayed or language impaired children.

In relation to the third comment, my aim was to study language delay, its differential prevalence in PT and FT children and those variables which may predict language delay at different ages. The use of actual scores as the dependent variable would change the aims of this study (I have done it, however, in other studies). In any case, mean scores and SD and ANOVA comparisons on the raw scores are given in Table 2.

In relation to your final comment, two new paragraphs have been included which introduce a more relativistic position in relation to the non-effect of gestational age and the practically inexistent differences between PT and FT children (see lines 499-500 and 666-667).

Reviewer 2 Report

Worthwile study with important findings that contribute to the advancement of knowledge. There are, however, a few points that need to be addressed before publication. 

General comments/main issues:

Section 2.1. Where were the participants recruted (nothing too specific, just country and, possibly, region would suffice). You mention data collection instruments in Galician later on, were all the participants recruted in Galicia, then?
Section 2.2. You mention instruments in Galician and in Spanish. Were the participants tested in both languages, only in Gallego, only in Spanish? 
Also, along this lines, as the study was carried out in a bilingual region, it is essential to report on the bilingualism status of your participants. If you tested your participants in Spanish, were you able to determine that there were Spanish dominant? Were there balanced bilingual or Galician dominant participants? What were the languages spoken at home? 
All this information is crucial given the scope of your study and getting a clear picture of the bilingualism status of your participants is particularly important, as it is known that bilingual speakers may show an apparent delay in the early stages of language development when compared to monolingual speakers (even though they catch up fast, the apparent delay usually shows up when only one of the languages is considered, but not so much when both languages are considered, etc.). This is especially important for the data collections at 22 and 30 months. 
It is very well possible that all participants have similar profiles in terms of exposure to the two languages, but it is important to clarify these issues to demonstrate the validity of the results. 

Throughout the manuscript:  there are seeral cases of double spacing between words.

In general, the text is well orgnized and clearly written, but the style doesn't always sound natural and there are several odd sentences scattered throughout the manuscript.  Proofreading by a native speaker would improve the quality of the paper.

Some acronyms (e.g. IVH, NICU...) are introduced without the corresponding expanded meaning, which often appears later in the text. 

Specific comments/minor issues: 

line 35-37, there is a lack of consistency when indicating GA (36 to 34 weeks... shouldn't it be 34-36 weks?)

line 50. Incomplete information. When do they show these differences with respect to FT? For how long?

line 80 (and also line 663). There is no Appendix A...
line 106. "which are not coincident" odd wording, please consider rephrasing.

line 131. "admitted"... "included" would sound better, I think.

line 228-233. The goals of the study could be presented earlier

line 334. "(enter method)" There seems to be missing information.

line 394. So, GA of 31 weeks or below show similar proportions of LD to FT, i.e. lower than later PT participants? I find that result rather surprising, and wonder what could explain it. 

line 520-521 "this result confirms that the use of gestures seems to have an effect on language development in the short term." I'm not sure this claim is accurate, it could very well be that the use of gestures at 10 months is a good predictor of language development at 22 months. So, not that gestures have an effect on language development, but rather that they are a predictor/indicator of it. Thus, I think it would be safer to phrase it as an indicator, a possible predictor or as an apparent correlation.

line 527 "sensibility" or sensitivity? (same thing on lines 542-543 and 594)

line 614 "amazing" - inapropriate

Author Response

RESPONSE TO REVIEWER 2

I very much appreciate your comments and suggestions.

In relation to your comments on Section 2.1, a new paragraph has been introduced giving the required information (see lines 240-241 of the corrected version).

Regarding your comments on section 2.2, a new paragraph has been introduced on page 7 (lines 319-322), trying to explain the use of the tests in different languages (Galician and Spanish). A longer explanation would lengthen the paper and would, probably, be out of place taking into account the aims of the study. It is certainly true than when bilingual children’s vocabulary is assessed in only one language, they may obtain lower results than monolingual children of the same language. However, when bilingual children are assessed with two tests for each language and a total or composite score of vocabulary is obtained, they usually get similar results to monolingual children [1].  This is particularly clear for bilingual Spanish-English children (two distant languages); however, with two languages which are very close, as Galician and Spanish are, the situation is different. In the case of the IDHC (the Galician version of the CDI), around 65% of the words in Galician are practically the same as in Spanish. Small differences may exist between a few words, which are difficult to appreciate given the pronunciation of the children (for instance cabalo and caballo ‘horse` in Galician and Spanish, respectively). To reinforce this, in a comparative study we performed with monolingual Portuguese children (another language close to Galician), the Galician children did not obtain lower results than the Portuguese children of the same age [2].

I tried to eliminate all the double spacing between words.

A native speaker has checked the text.

The use of acronyms has been checked (see track changes).

The lack of consistency appreciated in lines 35-37 has been corrected.

In line 50 (now 52) a short comment has been added to complete the information. In any case, this was a general statement of secondary relevance which only tried to show that difficulties also included grammar.

Former line 80 (now 82): as far as I understand in accordance with the norms of publication in this journal, Appendix B is suitable for the information I provide in Table A1, not Appendix A. Therefore, there is no Appendix A.

Line 106 (now 108), "which are not coincident" has been changed for “which have provided diverging results”.

Line 131 (now 133), the suggested change has been introduced.

Line 228-233. The goals of the study were presented at the end of the introduction and before the Methods section, as usual. However, in lines 61-65 (formerly, lines 59-63) the focus of the study is described.

Line 334 (now 347), “enter method”. The method used for the logistic regression analyses has been the “enter method” (default method in the SPSS), in which the investigator introduces the independent variables simultaneously, different from the backward or forward methods, in which the variables are automatically dropped by the program from a list. I do not consider it necessary to give more information on this in the text.

Line 394 (now lines 407-409). I agree with you that this is a surprising result. I commented on this result in the discussion section (lines 490-495), although the explanation seems to be more complex.

Line 520-521 (now 541-542). Your suggestion is accurate, and the text has been changed.

Line 527 (and others). Sensibility is a mistake, and it has been changed to sensitivity everywhere. Thank you for the warning.

Line 614 (now 639) "amazing” has been changed for “surprising (and relatively unexpected) result”

References.

  1. Core, C.; Hoff, E.; Rumiche, R.; Señor, M., Total and Conceptual Vocabulary in Spanish-English Bilinguals From 22 to 30 Months: Implications for Assessment. Journal of speech, language, and hearing research: JSLHR 2013, 56.
  2. Viana, F. L.; Pérez-Pereira, M.; Cadime, I.; Silva, C.; Santos, S.; Ribeiro, I., Lexical, morphological and syntactic development in toddlers between 16 and 30 months old: A comparison across European Portuguese and Galician. First Language 2017, 37 (3), 285-300.